# Study Protocol for a Prospective Self-Controlled Trial on Success in Meeting Comprehensive Genomic Profiling Analysis Criteria for Specimens Obtained by Endoscopic Ultrasound-Guided Tissue Acquisition Using a 19G Needle from Primary and Metastatic Lesions in Pancreatic Cancer with Metastatic Lesions: The PRIMATE Study

**DOI:** 10.3390/diseases12080182

**Published:** 2024-08-12

**Authors:** Kotaro Takeshita, Susumu Hijioka, Yoshikuni Nagashio, Hidenobu Hara, Daiki Agarie, Yuki Kawasaki, Tetsuro Takasaki, Shin Yagi, Yuya Hagiwara, Kohei Okamoto, Daiki Yamashige, Soma Fukuda, Masaru Kuwada, Yasuhiro Komori, Mao Okada, Yuta Maruki, Chigusa Morizane, Hideki Ueno, Yasushi Yatabe, Takuji Okusaka

**Affiliations:** 1Department of Hepatobiliary and Pancreatic Oncology, National Cancer Center Hospital, Tokyo 104-0045, Japan; k.takeshita@tane.or.jp (K.T.); yonagash@ncc.go.jp (Y.N.); hihar@ncc.go.jp (H.H.); dagarie@ncc.go.jp (D.A.); yukikawa@ncc.go.jp (Y.K.); ttakasak@ncc.go.jp (T.T.); shyag@ncc.go.jp (S.Y.); yuhagiw2@ncc.go.jp (Y.H.); kookamot@ncc.go.jp (K.O.); dyamashi@ncc.go.jp (D.Y.); sofukud@ncc.go.jp (S.F.); mkuwada@ncc.go.jp (M.K.); yakomor@ncc.go.jp (Y.K.); maookada@ncc.go.jp (M.O.); ymaruki@ncc.go.jp (Y.M.); cmorizan@ncc.go.jp (C.M.); hiueno@ncc.go.jp (H.U.); tokusaka@ncc.go.jp (T.O.); 2Department of Gastroenterology, Tane General Hospital, Osaka 550-0025, Japan; 3Department of Diagnostic Pathology, National Cancer Center Hospital, Tokyo 104-0045, Japan; yyatabe@ncc.go.jp

**Keywords:** endoscopic ultrasound-guided tissue acquisition, comprehensive genomic profiling, pancreatic cancer, metastatic lesion, fine-needle biopsy

## Abstract

**Simple Summary:**

Although comprehensive genomic profiling (CGP) has attracted attention in unresectable pancreatic cancer with metastasis, the success rate of CGP in specimens obtained by endoscopic ultrasound-guided tissue acquisition (EUS–TA) is not high. This study is expected to improve the success rate of CGP of unresectable pancreatic cancer with metastasis in the future by clarifying whether the optimal target of EUS–TA is primary or metastatic lesions.

**Abstract:**

EUS–TA in unresectable pancreatic cancer requires not only a tissue diagnosis but also tissue collection in anticipation of comprehensive genomic profiling. However, the optimal puncture target remains controversial. Therefore, the Primary and Metastatic Lesions in Pancreatic Cancer (PRIMATE) study was designed to clarify the optimal target by comparing the success rates for meeting OncoGuide NCC Oncopanel (NOP) analysis criteria on pre-check primary and metastatic lesion specimens obtained during the same EUS–TA session in patients with invasive pancreatic ductal adenocarcinoma. In this ongoing prospective study, two specimens, each from primary and metastatic lesions, are obtained by EUS–TA (typically using a 19G fine-needle biopsy needle) in patients with invasive pancreatic ductal adenocarcinoma. The primary endpoint is the proportion of EUS–TA specimens that meet NOP analysis criteria during pre-check (i.e., tumor cellularity of ≥20% and a tissue area of ≥4 mm^2^), which are then compared between primary and metastatic lesions. This study has been approved by the National Cancer Center Institutional Review Board (Research No. 2022-168). The results of this study will be reported at an international conference and published in an international peer-reviewed journal. The trial registration number is UMIN 000048966.

## 1. Introduction

In recent years, cancer genomic medicine has been implemented in clinical practice, and comprehensive genomic profiling (CGP) using next-generation sequencers (NGSs), which can detect a large number of gene sequences in tumor tissue specimens, has become available [1,2,3,4,5,6,7,8,9,10,11]. In pancreatic cancer, CGP has made it possible to select treatment options, such as molecularly targeted drugs and immune checkpoint inhibitors, that are suited to the patient, with new treatment options expected in the future [12,13,14,15,16,17,18,19,20,21]. Therefore, endoscopic ultrasound-guided tissue acquisition (EUS–TA) in unresectable pancreatic cancer requires not only a tissue diagnosis but also tissue collection in anticipation of CGP [22].

CGP systems currently approved in Japan include the OncoGuide NCC Oncopanel System (NOP; Sysmex Corporation, Hyogo, Japan) and Foundation One CDx Cancer Genome Profile (Fone; Foundation Medicine, Cambridge, MA, USA). The specimen criteria for the NOP system are tumor cellularity of ≥20% and a tissue area of ≥4 mm^2^, and that for the Fone system are tumor cellularity of ≥20% and a tissue area of ≥25 mm^2^ (Table 1).

In clinical settings, a pathologist evaluates the quality of the tissue specimen obtained by EUS–TA (during pre-check), and only cases that meet the criteria are subjected to CGP analysis (Figure 1). As pancreatic cancer is representative of low-cellularity tumors with abundant stromal components, the tumor cell content is reported to be 5–20%; accordingly, a greater amount of adequate tissue for CGP is required [23].

Two prospective studies have examined the proportion of successful CGP analyses of pancreatic tumor specimens. Carrera et al. reported a success rate of 97% (32/33) for CGP using NGS of cryopreserved tissue samples obtained by fine-needle biopsy (FNB) with a 22G needle [24]. Hisada et al. analyzed the pre-check results of EUS–TA specimens obtained by EUS–FNB with a 19G needle from a single lesion in patients with unresectable pancreatic cancer with metastasis (UR–M) and found that 63.6% (21/33) of specimens met NOP criteria [25]. Additionally, several retrospective studies have examined the success rate of genomic analysis of the pancreatic cancer specimens obtained using an FNB needle [24,26,27,28,29,30], including two analyses that focused exclusively on invasive pancreatic ductal carcinoma. Elhanafi et al. performed a retrospective study of EUS–TA in 167 patients with pancreatic cancer and found that 70.1% of specimens met NGS analysis criteria [29]. However, this study utilized a low threshold in the criterion for tumor cellularity (≥10%) in the pre-check. Park et al. performed a retrospective study of 190 patients with invasive pancreatic ductal adenocarcinoma and reported an NGS analysis success rate of 57.4% [30]. However, this study utilized a lower threshold for DNA content (≥50 ng) than that required for the NOP system. Thus, the reported percentages of specimens meeting NGS analysis criteria show considerable variation (57.4–100%), likely because of differences in outcome measures (percentage of successful analyses rather than that of the pre-check), specimen preservation methods, patient population (invasive pancreatic ductal adenocarcinoma with or without other cancer types), and specimen criteria (unspecified or lower criteria than that required for the NOP system).

The relationship between the EUS–TA technique and the percentage of successful CGP analyses has also been evaluated. Park et al. reported that the puncture needle diameter (22G or 19G) and tumor site (body–tail), but not needle type (fine needle aspiration or FNB), contributed to a successful CGP analysis of primary pancreatic specimens collected using EUS–TA [30]. Larson et al. reported that large FNB needle diameter (19G, *n* = 2; 22G, *n* = 41; and 25G, *n* = 2) was associated with a high success rate for meeting NGS analysis criteria (*p* = 0.05) [28]. Therefore, when performing EUS–TA with an FNB needle for CGP, a large puncture needle diameter, with a large amount of tissue sampled, is recommended to achieve a high analysis success rate.

However, the optimal puncture target remains controversial. Larson et al. compared the proportion of specimens that met CGP criteria between those obtained by EUS–TA for primary pancreatic lesions or percutaneous biopsies for metastatic lesions and found a significantly higher success rate for metastatic lesions than for primary lesions, despite the difference in sampling modalities (*p* = 0.036) [28]. Ikeda et al. reported that, overall, 39.2% (60/153) of pancreatic specimens obtained with an FNB needle met NOP analysis criteria during pre-check; however, primary lesion specimens (37.1%, 53/143) were significantly less likely than metastatic lesion specimens (70%, 7/10) to meet analysis criteria (*p* = 0.049) [31]. Nevertheless, data on the proportion of metastatic lesion specimens that meet CGP analysis criteria remain insufficient, and it is not yet clear whether primary or metastatic lesions have a high success rate for CGP analysis of invasive pancreatic adenocarcinoma.

Therefore, we designed a study to clarify the appropriate target for EUS–TA when CGP is planned for pancreatic ductal adenocarcinoma by comparing the success rates for meeting NOP analysis criteria between primary and metastatic lesion specimens obtained during the same EUS–TA session in patients with invasive pancreatic ductal adenocarcinoma. The protocols are described herein.

## 2. Study Design

### 2.1. Patients, Ethics, and Results Dissemination

The study was approved by the National Cancer Center Institutional Review Board (Research No. 2022-168). Protocol version 1.2 was finally approved on 23 May 2023. A summary of the study, its progress, and main results will be made available in the clinical registry system of the University Hospital Medical Information Network (UMIN ID: 000048966). The results will be reported at an international conference and published in an international peer-reviewed journal. All patients will be briefed on the study, and informed consent will be obtained before enrollment.

### 2.2. Study Design

This is an ongoing single-center prospective self-controlled trial. The study flowchart is shown in Figure 2.

### 2.3. Endpoints

The primary endpoint is the proportion of EUS–TA specimens obtained from primary and metastatic lesions that meet NOP analysis criteria. Although the NOP product catalog indicates tumor cellularity of ≥20% and a tissue area of ≥16 mm^2^, in actual clinical practice, a tissue area of ≥4 mm^2^ provides adequate DNA content (≥200 ng) for analysis. Thus, specimens with tumor cellularity of ≥20% and a tissue area of ≥4 mm^2^ are considered to meet NOP analysis criteria.

The secondary endpoints are as follows: rate of patients with primary or metastatic lesion specimens that meet NOP analysis criteria; technical success rates of EUS–TA with a TopGain 19G needle in primary and metastatic lesions; technical success rates of EUS–TA (regardless of needle type) in primary and metastatic lesions; adverse event rate, details, and severity; sensitivity, specificity, and accuracy of the histological differentiation of malignant and benign lesions in primary or metastatic lesion specimens; rates of successful NOP analysis of primary and metastatic lesion specimens; rate of patients who receive a drug targeting detected genetic mutations; details of genetic mutations detected upon NOP analysis; tumor cellularity and tissue area of all EUS–TA sessions; tumor cellularity and tissue area according to the nCounter system for all EUS–TA sessions; rate of patients who do not meet NOP criteria for both primary and metastatic lesion specimens and underwent liquid biopsy; rate of the necessity of percutaneous biopsy because specimens do not meet NOP analysis criteria for primary or metastatic lesions; and DNA amount in cases analyzed for DNA amount.

### 2.4. Eligibility Criteria

Eligibility criteria are as follows: suspected invasive pancreatic ductal adenocarcinoma (or a confirmed diagnosis of invasive pancreatic ductal adenocarcinoma); metastatic lesions on pre-procedure imaging or prior EUS examination; aged older than 18 years; Class 0–2 on the Eastern Cooperative Oncology Group Performance Status scale; no gastrointestinal strictures in the pharynx or esophagus on pre-procedural imaging or prior endoscopy examination; pre-procedure imaging or EUS indicating that both primary and metastatic lesions can be safely punctured in EUS–TA with a 19G FNB needle, without interventions on blood vessels, other tumors, or other organs; no prior radiotherapy or chemotherapy for pancreatic cancer; able to withdraw from antithrombotic medication based on guideline criteria [32]; no specific bleeding risk (prothrombin time–international normalized ratio < 1.5 or platelet count > 50,000/L); and no more than a moderate volume of ascites (i.e., clear accumulation of ascites at the site of the possible puncture route for EUS–TA).

### 2.5. Exclusion Criteria

The exclusion criteria are as follows: two or more lesions in the pancreas suspected of pancreatic cancer; psychosis or psychiatric symptoms that interfere with daily life and make study participation difficult; and enrollment in the study deemed as inappropriate by a physician.

### 2.6. Trial Examination Procedures

Participants undergo EUS performed by a physician who has performed at least 100 cases of EUS–TA as a primary physician. Pancreatic primary and metastatic lesions are characterized, and the maximum lesion diameter is measured in B-mode, followed by confirmation that there are no intervening vessels in the puncture route in Doppler mode. EUS–TA is typically performed using a TopGain 19G needle (SonoTip TopGain; Medi-Globe, Achenmuhle, Germany). Negative pressure is applied using suction with a 20 mL syringe or the slow-pull method, and approximately 20 actuations per pass are performed. If possible, the fanning technique is performed during actuation, but it is not mandatory. When a lesion cannot be identified or the puncture line is technically difficult, a change in the puncture needle or physician is considered. If the puncture remains difficult, it is not forced, and the patient is considered a technical failure. If both primary and metastatic lesions are difficult to puncture, alternative biopsies, such as percutaneous biopsies, are considered. Changing the puncture needle for each lesion is not mandatory, but if the needle deteriorates, it is acceptable to use a new needle as needed. It is not specified whether primary or metastatic lesions should be punctured first, but the puncture is basically performed alternately on the primary and metastatic lesions.

For both primary and metastatic lesions, obtained specimens are placed in shale, and a portion of the specimen is cut and considered for rapid on-site evaluation (ROSE). However, ROSE is not mandatory. When ROSE is performed, the remaining specimens are not included in the study because of the small number of specimens. If malignant cells are not confirmed by ROSE, changes in the puncture site, target, and needle are considered; however, a diagnosis of malignancy may be made by histological examination in some cases. Therefore, even if malignant cells cannot be confirmed using ROSE, additional puncturing, without changing the site or puncture method, is considered acceptable. All collected specimens, including residual specimens after ROSE, are individually preserved in formalin bottles for histological evaluation.

The procedure is terminated once two specimens each are obtained from primary and metastatic lesions, after confirming by EUS and endoscopy that there are no findings suggestive of hemorrhage. If bleeding is suspected, endoscopic hemostasis or endovascular treatment will be considered.

Specimens obtained by EUS–TA are thinly sliced after formalin fixation and stained with hematoxylin and eosin. Specimens, excluding those from the ROSE session, are checked to determine whether they meet NOP analysis criteria. The sample volume required for NOP analysis is five 10 μm thick sections, with tumor cellularity of ≥20% and total DNA content of ≥200 ng per slide. A tissue area of approximately 16 mm^2^ per slide is recommended; however, a tissue area of 4 mm^2^ or more yields > 200 ng of total DNA. Therefore, we consider specimens with tumor cellularity of ≥20% and a tissue area of ≥4 mm^2^ as successfully meeting NOP analysis criteria. The pathological evaluation is performed by two pathologists, and any discrepancies in the measurements are resolved by discussion. In this evaluation, tumor cellularity is evaluated at five levels (>80%, 60–80%, 40–60%, 20–40%, and <20%), and tissue area is evaluated at four levels (≥25, 16–25, 4–16, and <4 mm^2^).

### 2.7. Sample Size Determination and Statistical Analysis

The proportion of metastatic lesion specimens that meet NOP analysis criteria is expected to be approximately 70%. The proportion of primary lesions that meet NOP analysis criteria is likely to be lower than the 63.6% reported by Hisada et al. because of the reduced number of passes in the PRIMATE study. In addition, in our previous retrospective study, the proportion of primary tumor specimens obtained by EUS–TA with a 19G needle meeting criteria on pre-check was 53.8%, with a median number of punctures of four. Therefore, the proportion of primary lesion specimens that meet NOP analysis criteria in the present study is expected to be approximately 45%. Accordingly, the minimum sample size required for a Fisher’s exact test is 56 patients, assuming a one-sided significance level of 0.05 and a power of 80%. Assuming that approximately 10% of registered patients will experience difficulty in puncturing, a sample size of 61 patients is planned.

The proportion of EUS–TA specimens meeting NOP analysis criteria will be compared between primary and metastatic lesions using Fisher’s exact test. Statistical significance is set at *p* < 0.05.

## 3. Discussion

This study will play an important role in determining the target of EUS–TA in pancreatic cancer with distant metastases when CGP analysis is considered. However, there are several limitations. First, the study will not randomize the initial lesion to be punctured and will allow for the same needle to be used to puncture the primary and metastatic lesions, which may have an impact on EUS–TA as the needle tip gradually deteriorates over time, making specimen collection more difficult. However, the primary and metastatic lesions will be punctured alternately after the first lesion is punctured. Changing the puncture needle at the discretion of the physician when they sense needle deterioration will be allowed. These make this the least confounding factor.

The second is the EUS–TA method. The usefulness of wet suction has been reported in recent years [33,34,35,36,37,38,39,40,41,42,43], with better quality specimens obtained compared with those obtained by dry suction. However, in the meta-analysis, the specimen adequacy rate was similar to the slow-pull method and the risk ratio was 1.02 (0.98–1.07) [36]. Moreover, there is no report to date that wet suction is superior to dry suction or the slow-pull method in terms of specimen volume when genomic testing is taken into consideration. In terms of specimen volume with a view of CGP analysis, whether wet or dry suction is better may be an issue for future studies. The primary endpoint of this study is the comparison of the rate of primary and metastatic lesions that met the criteria of the NOP analysis, and as long as the same slow-pull method of negative pressure is applied to both groups, the fact that wet suction is not applied will not affect the primary endpoint.

Finally, double blinding is not possible. The physician naturally knows whether to puncture the primary lesion or the metastatic lesion, which can be a confounding factor. However, despite these limitations, there have been few reports on whether primary lesions or metastatic lesions are more appropriate for collecting specimens for CGP analysis, and this study may provide significant evidence.

## 4. Conclusions

In clinical settings, the first choice for the target of EUS–TA in UR–M pancreatic ductal adenocarcinoma is the pancreatic primary lesion. However, the success rate for meeting CGP analysis criteria is not high in specimens acquired from the primary lesion because one characteristic of pancreatic ductal adenocarcinoma is low cellularity. If this study finds that the percentage of metastatic lesion specimens meeting NOP analysis criteria (trial examination) exceeds the percentage of primary lesion specimens meeting the criteria (standard examination), metastatic lesions should be recommended as the target of EUS–TA when considering CGP in UR–M pancreatic ductal adenocarcinoma. This may allow more patients with UR–M pancreatic ductal adenocarcinoma to benefit from CGP. It may be possible to eliminate the conventional two-step procedure of EUS–TA for the primary lesion for diagnosis and percutaneous liver biopsy of the metastatic lesion for CGP. With their incorporation into future guidelines, the results of this study may change the standard for selecting targets for EUS–TA in pancreatic ductal adenocarcinoma with metastatic lesions.

## Figures and Tables

**Figure 1 diseases-12-00182-f001:**
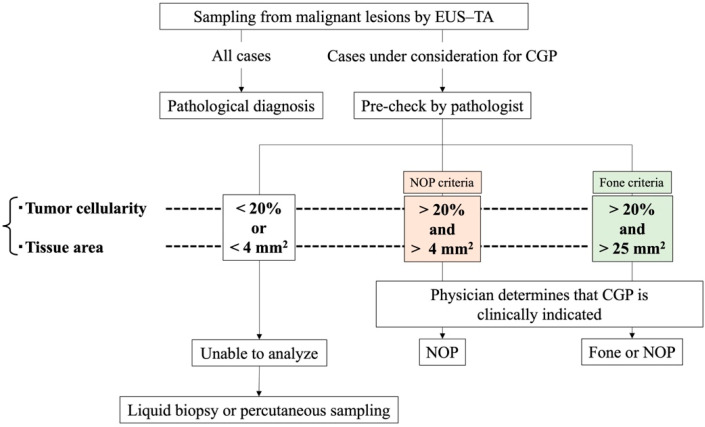
Flowchart of clinical CGP after EUS–TA. EUS–TA, endoscopic ultrasound-guided tissue acquisition; CGP, comprehensive genomic profiling; NOP, NCC Oncopanel; Fone; Foundation One CDx.

**Figure 2 diseases-12-00182-f002:**
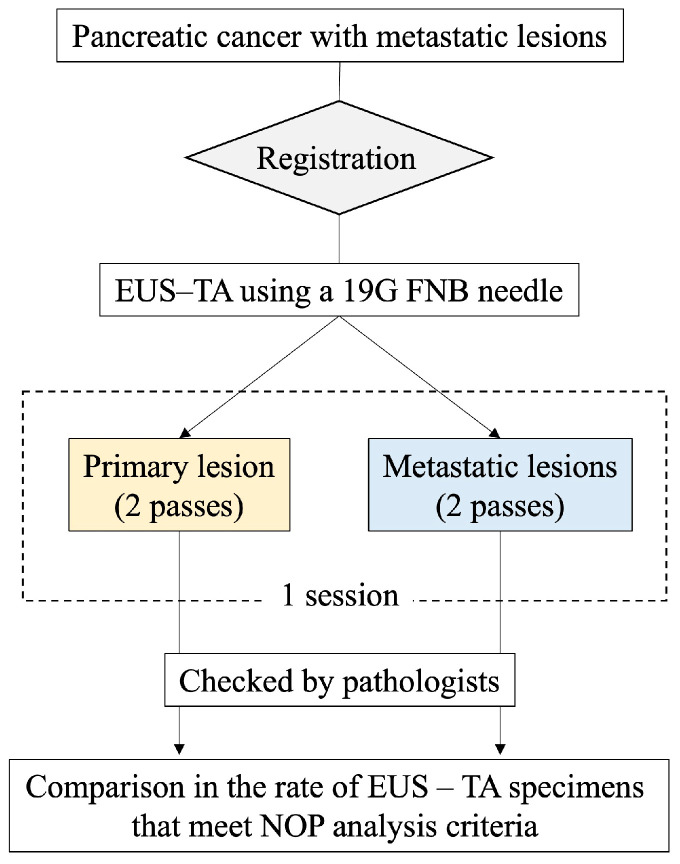
Registration flowchart for the PRIMATE study. EUS–TA, endoscopic ultrasound-guided tissue acquisition; FNB, fine needle biopsy; NOP, NCC Oncopanel.

**Table 1 diseases-12-00182-t001:** Differences between the NCC Oncopanel system and Foundation One CDx.

	NOP	Fone
Required tissue area	≥16 mm^2^ (4 mm^2^ is acceptable)	≥25 mm^2^
Required tumor cellularity	>20% DNA content (≥200 ng)	>20%

NOP, NCC Oncopanel; Fone; Foundation One CDx.

## Data Availability

The data presented in this study are available on request from the corresponding author.

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
