# Peer review of "Study Protocol for a Prospective Self-Controlled Trial on Success in Meeting Comprehensive Genomic Profiling Analysis Criteria for Specimens Obtained by Endoscopic Ultrasound-Guided Tissue Acquisition Using a 19G Needle from Primary and Metastatic Lesions in Pancreatic Cancer with Metastatic Lesions: The PRIMATE Study"

_diseases, 2024, doi:10.3390/diseases12080182_

Round 1
Reviewer 1 Report
Comments and Suggestions for Authors
General:
This is a report on an ongoing study protocol and its scheme and rationale for comprehensive genomic profiling analysis using tissue acquired during EUS. This report does not have a result. Therefore, not a research article.
Major issues:
This is not an original research study article or incomplete study with no result of testing and/or measured parameters with the identified population. Would wait for the study results, and the conclusion derived from these. Otherwise, it needs to be formatted as different type of paper all together.
Minor issues:
N/A
Author Response
Comments1: [This is not an original research study article or incomplete study with no result of testing and/or measured parameters with the identified population. Would wait for the study results, and the conclusion derived from these. Otherwise, it needs to be formatted as different type of paper all together.]
Response1: [Thank you very much for reviewing our paper. As you point out, this paper is a protocol paper and not an original research paper. We have revised the article type to "Protocol" accordingly. We apologize for the initial discrepancy.]
Reviewer 2 Report
Comments and Suggestions for Authors
In this paper, the authors report a protocol of an ongoing study conducted in a single center. The topic of the study is certainly important. Indeed, the main outcome is the rate of endoscopic ultrasound specimens acquired from the primary pancreatic cancer or metastatic lesion adequate for next-generation sequencing according to the NCC Oncopanel 54 System (NOP). Below are my comments:
1) Looking at the procedures described in the protocol, you will use a 19G Franseen needle.
You will use the dry suction or the slow-pull technique. However, a recent meta-analysis demonstrated that the wet suction provides a larger amount of tissue (PMID: 36657607). Please add this point as a study limitation and cite the abovementioned study.
2) Will you use the fanning technique? It is recommended that different parts of the tumor be sampled to increase the change of adequate tissue. Please clarify.
3) If I understand well, you will perform a total of 4 passes using the same needle. The first two will target the primary tumor, and the last 2 will target the metastasis. I think that there is a chance that the needle performance is lower during the last passes (due to reduced tip sharpening), thus impacting the primary outcome. Have you thought about sampling metastatic and primary tumors in a randomized order to reduce this bias? Or you could change the needle and use a new one when changing the sampling target. Please add this point to the discussion.
4) A discussion paragraph should be added, including the study’s limitations.
Author Response
Response to reviewer2
Thank you very much for reviewing our paper. We have provided point-by-point responses to your comments below and have revised the manuscript accordingly.
Comments1: [Looking at the procedures described in the protocol, you will use a 19G Franseen needle.
You will use the dry suction or the slow-pull technique. However, a recent meta-analysis demonstrated that the wet suction provides a larger amount of tissue (PMID: 36657607). Please add this point as a study limitation and cite the abovementioned study.]
Response1: [Thank you for your suggestion. As you point out, the usefulness of wet suction has been reported in recent years. However, the superiority of wet suction in terms of specimen volume has not been fully proven. In the meta-analysis you cite, the comparison of specimen adequacy rates between modified wet suction and the slow pull method showed a risk ratio of 1.02 (0.97–1.02), indicating that there is little difference. However, the primary endpoint of the study described in our paper is the comparison of the CGP analysis success rates between primary and metastatic lesion specimens. As long as the same method is used for both lesions, we believe that this does not affect the endpoint.
We have included the suction method as a topic for future research in the Discussion section as follows:
- Discussion
This study will play an important role in determining the target of EUS–TA in pancreatic cancer with distant metastases when CGP analysis is considered. However, there are several limitations. First, the study will not randomize the initial lesion to be punctured and will allow the same needle to be used to puncture the primary and metastatic lesions, which may have an impact on EUS–TA as the needle tip gradually deteriorates over time, making specimen collection more difficult. However, the primary and metastatic lesions will be punctured alternately after the first lesion is punctured. Changing the puncture needle at the discretion of the physician when they sense needle deterioration will be allowed. These make this the least confounding factor.
The second is the EUS–TA method. The usefulness of wet suction has been reported in recent years [33-43], with better quality specimens obtained compared with those obtained by dry suction. However, in the meta-analysis, the specimen adequacy rate was similar to the slow-pull method and the risk ratio was 1.02 (0.98–1.07) [36]. Moreover, there is no report to date that wet suction is superior to dry suction or the slow-pull method in terms of specimen volume when genomic testing is taken into consideration. In terms of specimen volume with a view of CGP analysis, whether wet or dry suction is better may be an issue for future studies. The primary endpoint of this study is the comparison of the rate of primary and metastatic lesions that met the criteria of the NOP analysis, and as long as the same slow-pull method of negative pressure is applied to both groups, the fact that wet suction is not applied will not affect the primary endpoint.
Finally, double-blinding is not possible. The physician naturally knows whether to puncture the primary lesion or the metastatic lesion, which can be a confounding factor. However, despite these limitations, there have been few reports on whether primary lesions or metastatic lesions are more appropriate for collecting specimens for CGP analysis, and this study may provide significant evidence.]
Comments2: [Will you use the fanning technique? It is recommended that different parts of the tumor be sampled to increase the change of adequate tissue. Please clarify.]
Response2: [Thank you for your comments. In our study protocol, the fanning technique is not specified. This is because fanning is not feasible for all primary and metastatic lesions. However, for tumors with internal heterogeneity, such as pancreatic cancer, the fanning technique is an important method. In our institution, we actively use the fanning technique whenever possible. We have added this information to the revised manuscript, as follows:
EUS–TA is typically performed using a TopGain 19G needle (SonoTip TopGain; Medi-Globe, Achenmuhle, Germany). Negative pressure is applied using suction with a 20-mL syringe or the slow-pull method, and approximately 20 actuations per pass are performed. If possible, the fanning technique is performed during actuation, but it is not mandatory.]
Comments3: [If I understand well, you will perform a total of 4 passes using the same needle. The first two will target the primary tumor, and the last 2 will target the metastasis. I think that there is a chance that the needle performance is lower during the last passes (due to reduced tip sharpening), thus impacting the primary outcome. Have you thought about sampling metastatic and primary tumors in a randomized order to reduce this bias? Or you could change the needle and use a new one when changing the sampling target. Please add this point to the discussion.]
Response3: [Thank you for pointing out this important issue. We apologize for the oversight in explaining the order of puncturing primary and metastatic lesions.
As you point out, the degradation of the FNB needle tip can be a concern with repeated punctures. In the study described in our paper, the order of puncturing is not specified in the protocol, allowing for either primary or metastatic lesions to be punctured first without any issues. Although the protocol does not mandate a specific order, in most cases, punctures are performed in the order of primary → metastatic → primary → metastatic.
While the protocol does not require changing needles between primary and metastatic punctures, it permits the use of a new needle if degradation is observed with repeated punctures. We have added this information to section 2.6 of the revised manuscript, as follows:
When a lesion cannot be identified or the puncture line is technically difficult, a change in the puncture needle or physician is considered. If the puncture remains difficult, it is not forced, and the patient is considered as a technical failure. If both primary and metastatic lesions are difficult to puncture, alternative biopsies, such as percutaneous biopsies, are considered. Changing the puncture needle for each lesion is not mandatory, but if the needle deteriorates, it is acceptable to use a new needle as needed. It is not specified whether primary or metastatic lesions should be punctured first, but the puncture is basically performed alternately on the primary and metastatic lesions.]
Comments4: [A discussion paragraph should be added, including the study’s limitations.]
Response4: [Thank you for your comment. We have updated the discussion as suggested:
- Discussion
This study will play an important role in determining the target of EUS–TA in pancreatic cancer with distant metastases when CGP analysis is considered. However, there are several limitations. First, the study will not randomize the initial lesion to be punctured and will allow the same needle to be used to puncture the primary and metastatic lesions, which may have an impact on EUS–TA as the needle tip gradually deteriorates over time, making specimen collection more difficult. However, the primary and metastatic lesions will be punctured alternately after the first lesion is punctured. Changing the puncture needle at the discretion of the physician when they sense needle deterioration will be allowed. These make this the least confounding factor.
The second is the EUS–TA method. The usefulness of wet suction has been reported in recent years [33-43], with better quality specimens obtained compared with those obtained by dry suction. However, in the meta-analysis, the specimen adequacy rate was similar to the slow-pull method and the risk ratio was 1.02 (0.98–1.07) [36]. Moreover, there is no report to date that wet suction is superior to dry suction or the slow-pull method in terms of specimen volume when genomic testing is taken into consideration. In terms of specimen volume with a view of CGP analysis, whether wet or dry suction is better may be an issue for future studies. The primary endpoint of this study is the comparison of the rate of primary and metastatic lesions that met the criteria of the NOP analysis, and as long as the same slow-pull method of negative pressure is applied to both groups, the fact that wet suction is not applied will not affect the primary endpoint.
Finally, double-blinding is not possible. The physician naturally knows whether to puncture the primary lesion or the metastatic lesion, which can be a confounding factor. However, despite these limitations, there have been few reports on whether primary lesions or metastatic lesions are more appropriate for collecting specimens for CGP analysis, and this study may provide significant evidence.]
Reviewer 3 Report
Comments and Suggestions for Authors
This is a very interesting study. I would like to see a study using 22G needles in addition to 19G needles.
Comments on the Quality of English LanguageMinor editing of English language required.
Author Response
Comments [Comments on the Quality of English Language
Minor editing of English language required.]
Response
Thank you very much for reviewing our paper. Although the English in the manuscript had already been reviewed by a native speaker, we have subjected it to an additional rigorous review.
Round 2
Reviewer 1 Report
Comments and Suggestions for Authors
No additional.
Reviewer 3 Report
Comments and Suggestions for Authors
No additional comments.